# CoCoIns: Consistent Subject Generation via Contrastive Instantiated Concepts

**Lee Hsin-Ying**                                      *hlee307@ucmerced.edu*
*University of California, Merced*

**Kelvin C.K. Chan**                                   *kelvinckchan@google.com*
*Google DeepMind*

**Ming-Hsuan Yang**                                    *mhyang@ucmerced.edu*
*University of California, Merced*

**Reviewed on OpenReview:** *https://openreview.net/forum?id=fPZ7DNlOSn*

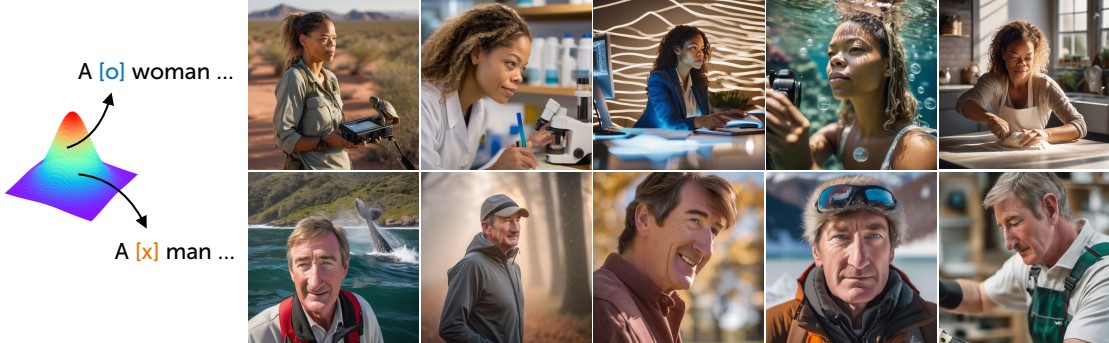

Figure 1. **Contrastive Concept Instantiation (CoCoIns)** is a generation framework that achieves subject consistency across multiple independent generations without fine-tuning or reference images. Unlike prior work that requires customization fine-tuning, adopts an additional encoder for references, or generates images in batches, CoCoIns creates *instances of concepts* with unique associations connecting latent codes to subject instances. Given a latent code (o/x), CoCoIns converts it into a pseudo-word ([o]/[x]) that determines the appearance of a subject concept. By reusing the same code, users can consistently generate the same subject instances across generations.

## Abstract

While text-to-image generative models can synthesize diverse and faithful content, subject variation across multiple generations limits their application to long-form content generation. Existing approaches require time-consuming fine-tuning, reference images for all subjects, or access to previously generated content. We introduce Contrastive Concept Instantiation (CoCoIns), a framework that effectively synthesizes consistent subjects across multiple independent generations. The framework consists of a generative model and a mapping network that transforms input latent codes into pseudo-words associated with specific concept instances. Users can generate consistent subjects by reusing the same latent codes. To construct such associations, we propose a contrastive learning approach that trains the network to distinguish between different combinations of prompts and latent codes. Extensive evaluations on human faces with a single subject show that CoCoIns performs comparably to existing methods while maintaining greater flexibility. We also demonstrate the potential for extending CoCoIns to multiple subjects and other object categories. The source code and model weights are available at https://contrastive-concept-instantiation.github.io.

# 1 Introduction

Text-to-image generation has made remarkable advances (Rombach et al., 2022; Saharia et al., 2022; Podell et al., 2023), opening up numerous downstream applications, including editing and style transfer. Among these applications, maintaining subject consistency has been a long-standing challenge in long-form content creation, including storytelling (Li et al., 2019), comics (Wu et al., 2024a), and movie generation (Tulyakov et al., 2018; Polyak et al., 2025). These applications consist of sequences of images and clips, where consistent characters and objects facilitate recognizing subjects across scenes and following narratives.

While numerous approaches have been explored to ensure subject consistency, they are often labor-intensive or time-consuming. One straightforward approach is to gather all existing generations and manually swap generated subjects with reference subjects using face swapping (Nirkin et al., 2019; Bitouk et al., 2008). Another direction is to customize generators by optimizing virtual word tokens or fine-tuning model weights to represent reference subjects and produce new generations (Gal et al., 2023a; Ruiz et al., 2022). To reduce the overhead of tuning generators, recent methods (Wei et al., 2023; Ye et al., 2023) incorporate additional encoders that convert references into representations. However, these methods still require users to prepare references for all subjects.

In contrast to addressing each generation individually, one can generate all images in a batch, allowing samples within the batch to interact and achieve consistency (Tewel et al., 2024; Zhou et al., 2024; Liu et al., 2025). Specifically, the target prompts are merged into the same batch, and the latents of all samples are processed together, allowing subjects within the batch to converge toward a similar appearance. Although these approaches achieve promising results, they require storing generated outputs to recreate the same subjects in the future.

We propose a generation framework that maintains subject consistency across individual generations without manual swapping, fine-tuning, or preparing references. Building such a framework presents numerous challenges. While we aim to generate a consistent subject appearance from a concept, preserving diversity among all instances of the concept remains important. Since the generator is already trained and exhibits high diversity and generalizability, we need to strike a balance between *minimizing variation* among the same subject instances across individual generations while *maintaining diversity* between different instances. Additionally, collecting large-scale, high-quality data organized by subjects is challenging. Training the generator directly on annotated subjects in low-quality datasets could hamper both output quality and diversity.

To minimize variation among instances of the same subject while preserving diversity among different instances, we introduce a latent space to model the distribution of instances for each concept. The proposed method is motivated by common practices in natural and programming languages. If a user provides sufficient descriptions that encompass every intricate aspect of a concept, the generator may be able to consistently output the same appearance. Although covering comprehensive details is impractical with the limited vocabulary of human language, prior work on customizing generative models has shown the efficacy of pseudo-words (Gal et al., 2023a; Ruiz et al., 2022), which can convey essential information to represent particular subject instances. Our framework is built upon *instantiating concepts* (Anderson et al., 1976; Dershowitz, 1985) via pseudo-words. We associate codes in the latent space with specific concept instances in the output space, as if creating instances identified by latent codes. These latent codes are embeddings sampled from the space, taken as input by the generation framework, and transformed into pseudo-words that guide the generator to synthesize specific instances.

To establish the association between input latent codes and output subject instances, we develop a lightweight mapping network that converts a latent code into a pseudo-word, which is then combined with a concept token to represent a specific instance of the concept. We then develop a contrastive learning strategy to train the mapping network. Instead of relying on subject annotations, the model learns to differentiate latent codes by comparing its own outputs generated from various combinations of prompts and latent codes. This self-supervised paradigm enables scalability and generalizability while avoiding the need to learn directly from limited data.

We refer to our generation framework as ***Contrastive Concept Instantiation (CoCoIns)***. As illustrated in Figure 1, given a concept in a prompt indicating a subject (*e.g.* woman), the framework creates an instance of that concept by transforming a sampled latent code into a pseudo-word (*e.g.* [o] in the example prompt)

that describes the concept. Each latent code and its transformed pseudo-word is uniquely tied to a specific instance and can be utilized for future generations. Different latent codes yield different instances, showcasing the preserved output diversity.

We conduct experiments on human images and perform systematic evaluations, including generating portrait photographs and free-form images. We achieve favorable subject consistency and prompt fidelity compared to batch-generation approaches. We also demonstrate early success in extending our approach to multi-subject and general concepts. The main contributions of this work are:

- To the best of our knowledge, we propose the first subject-consistent generation framework for multiple independent generations without fine-tuning or encoding references.
- We develop a contrastive learning method that avoids reliance on limited subject annotations while preserving output quality and diversity.
- We perform extensive evaluations and demonstrate favorable performance compared to approaches that require time-consuming fine-tuning or batch generation.

## 2  Related Work

**Subject-Driven Generation.** One approach to subject consistency is based on subject-driven generation, which aims to generate customized content according to user-provided input. By learning new tokens (Gal et al., 2023a; Voynov et al., 2023; Tewel et al., 2023) or model weights (Ruiz et al., 2022; Kumari et al., 2023; Han et al., 2023; Ruiz et al., 2024), pretrained generative models can be customized to produce outputs based on specific references. Textual Inversion (Gal et al., 2023a) learns virtual tokens that capture subject information inverted from reference images. DreamBooth (Ruiz et al., 2022) fine-tunes the parameters of pretrained models and learns unique identifiers that represent references. While subject consistency can be achieved by customizing each target concept with user-provided references, these methods are time-consuming as they require fine-tuning for every subject.

To reduce the time and computational cost of tuning-based methods, another line of research incorporates additional encoders to obtain representations from reference images, which generative models take as conditions via augmented prompt embeddings (Wei et al., 2023; Shi et al., 2024; Xiao et al., 2024; Li et al., 2023; Wang et al., 2024a; He et al., 2024; Chen et al., 2023; Gal et al., 2023b; Avrahami et al., 2023), self-attention (Ding et al., 2024; Wang et al., 2024b), or cross-attention (Wei et al., 2023; Shi et al., 2024; Jia et al., 2023; Ye et al., 2023; Wang et al., 2024b; 2025). In addition, some approaches (Valevski et al., 2023; Wang et al., 2024d; Li et al., 2024; Peng et al., 2024; Papantoniou et al., 2024; Wu et al., 2024b; Yue et al., 2024) focus solely on specific domains, *e.g.* human faces, in exchange for adopting more powerful encoders dedicated to those domains, *e.g.* face recognition models (Deng et al., 2022). While encoders ease the tuning process, users still need to prepare reference images for all target concepts. Designing specific mechanisms to insert reference features into generative models is also necessary. In contrast, our approach operates in the text embedding space, offering the potential to be applied to a wide range of text-based generative models.

**Subject-Consistent Generation.** While subject-driven generation produces new generations featuring the same subjects as references, another line of work focuses on generating a set of images with consistent subjects from a set of prompts. The Chosen One (Avrahami et al., 2024) iteratively selects a cluster of similar images and fine-tunes the model using that cluster. Consistory (Tewel et al., 2024), JeDi (Zeng et al., 2024), and StoryDiffusion (Zhou et al., 2024) utilize the self-attention features of all samples in a batch. However, these approaches are less flexible as they require access to other samples or features when performing new generations. 1Prompt1Story (Liu et al., 2025) consolidates all prompts into a single lengthy prompt in a specific format, where multiple context settings for a subject follow a single description of that subject. This format limits the expressiveness of the prompts. In contrast, we achieve subject consistency with greater flexibility by treating each generation independently while retaining complete prompts.

**Storytelling.** Generating coherent stories (Li et al., 2019) requires maintaining character consistency over time. Some approaches (Rahman et al., 2023; Pan et al., 2024; Liu et al., 2024; Shen et al., 2025; Maharana et al., 2022) utilize cross-attention to access information from previous frames and prompts. Others introduce a memory bank (Maharana et al., 2021), perform auxiliary foreground segmentation (Song et al., 2020), or

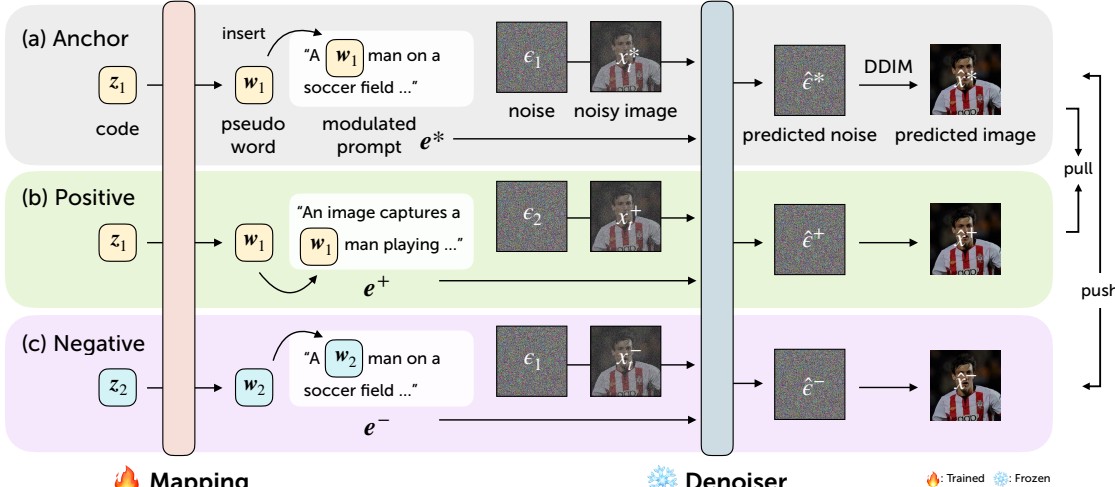

Figure 2. **Overview of Contrastive Concept Instantiation (CoCoIns)**. We develop a contrastive learning approach to build associations between input latent codes and concept instances. For each training image, we generate two descriptions and randomly sample two latent codes $z_1$ and $z_2$. The mapping network transforms the latent codes into pseudo-words $w_1$ and $w_2$. We then construct a triplet of combinations of descriptions and latent codes. We build (a) an anchor sample with description embedding $e^*$ modulated by inserting $w_1$ before the target concept token, (b) a positive sample $e^+$ with a similar description modulated with $w_1$, and (c) a negative sample $e^-$ with the same prompt as the anchor but modulated with a different pseudo-word $w_2$. The network is trained with a triplet loss to differentiate approximated images $\hat{x}^*$, $\hat{x}^+$, and $\hat{x}^-$ from denoiser predictions $\hat{\epsilon}^*$, $\hat{\epsilon}^+$, and $\hat{\epsilon}^-$.

generate particular reference characters (Gong et al., 2023; Shen & Elhoseiny, 2023; Wang et al., 2024e). These methods often involve training on storytelling datasets and focus on generating complete stories or continuing existing ones. Our approach emphasizes equipping existing generative models with the ability to maintain subject consistency.

**StyleGAN.** Our generation framework shares similar insights with StyleGAN (Karras et al., 2019; 2020). Both methods use a mapping network to transform input latents into an intermediate and more disentangled latent space. The space in StyleGAN enables better control over generated image attributes by modulating the generator through adaptive instance normalization (Huang & Belongie, 2017). In our framework, the intermediate latents operate in the same space as text embeddings, enabling better manipulation of subject appearances via text conditions. CharacterFactory (Wang et al., 2024c) also learns a mapping network that maps random noise to virtual tokens associated with human names in a celebrity dataset. These virtual tokens can produce consistent identities but lack control over semantics and attributes.

## 3 Methodology

Our goal is to maintain subject consistency across individual generations without time-consuming fine-tuning or labor-intensive reference collection. We introduce *Contrastive Concept Instantiation* (CoCoIns), a generation framework that models concept instances in a latent space and uniquely associates latent codes with output concept instances via contrastive learning. We introduce the base text-to-image model in Section 3.1, then describe the framework in Section 3.2 and the contrastive learning strategy in Section 3.3.

### 3.1 Text-to-Image Diffusion Models

We explore subject consistency in the context of text-to-image generation and base our approach on a latent diffusion model (Rombach et al., 2022; Podell et al., 2023). We utilize a pretrained text-to-image model comprising an autoencoder (Kingma & Welling, 2014), a text encoder (Radford et al., 2021), and a denoiser. Given an image $I$ and a prompt $P$, we obtain the latent image representation $x$ and prompt embedding $e$

via the autoencoder and text encoder, respectively. The denoiser $\epsilon_\theta$ reverses the diffusion process:

$$\boldsymbol{x}_t = \sqrt{\alpha_t}\boldsymbol{x} + \sqrt{1 - \alpha_t}\epsilon, \quad \epsilon \sim \mathcal{N}(\boldsymbol{0}, \boldsymbol{I}), \tag{1}$$

where $\alpha_{1:T} \in (0, 1]^T$ is a decreasing sequence, by predicting $\hat{\epsilon}$ from the noisy image $\boldsymbol{x}_t$, prompt $\boldsymbol{e}$, and timestep $t$:

$$\hat{\epsilon} = \epsilon_\theta(\boldsymbol{x}_t, \boldsymbol{e}, t). \tag{2}$$

## 3.2 Instantiating Concepts

To generate consistent instances of a concept, we model the distribution of instances in a latent space and associate latent codes in the space with concept instances in output images. As illustrated in Figure 2, a mapping network takes a latent code as input and produces a pseudo-word, which conveys the necessary descriptive details to create a particular concept instance. The framework thus achieves subject consistency by generating the same subject with a fixed latent code across multiple generations.

The proposed mapping network transforms a latent code into a virtual word token, which is then inserted into the prompt embedding and guides generation alongside other words. Let $\boldsymbol{e} \in \mathbb{R}^{s \times d}$ denote the embedding of a prompt $P$ obtained via dictionary lookup, where $s$ is the sequence length. The concept token that a user wants to generate consistently (*e.g. man* in Figure 2) is at location $i$. Given a latent code $\boldsymbol{z} \in \mathbb{R}^c$, the mapping network $f : \mathbb{R}^c \to \mathbb{R}^d$ produces a pseudo-word embedding $\boldsymbol{w} \in \mathbb{R}^d$ that represents an instance:

$$\boldsymbol{w} = f(\boldsymbol{z}), \quad \boldsymbol{z} \sim \mathcal{N}(\boldsymbol{0}, \boldsymbol{I}). \tag{3}$$

Then we insert the output $\boldsymbol{w}$ into the prompt embedding $\boldsymbol{e}$ at location $i$ before the concept token and obtain the modulated prompt embedding $\hat{\boldsymbol{e}}$:

$$\hat{\boldsymbol{e}} = \texttt{insert}(\boldsymbol{e}, \boldsymbol{w}, i), \tag{4}$$

where `insert` denotes the insertion operation. The modulated prompt embedding $\hat{\boldsymbol{e}}$ is further encoded by the text encoder and serves as the text condition during generation.

## 3.3 Contrastive Association

We aim to establish unique associations between input latent codes and pseudo-words that represent visual instances in the output images, such that a latent code can be reused to generate the same concept instance.

A naïve way is to train the network $f$ to synthesize subjects from a dataset with identity annotations, such as face recognition datasets (Huang et al., 2008; Liu et al., 2015). However, we empirically find that training with common noise prediction (Ho et al., 2020) often compromises the generalizability and output quality of the generator, as these datasets are typically collected from data domains that are much narrower than the pretraining data of the generator. The network learns to synthesize subjects from these datasets, but it may overfit to the limited distributions. Thus, we develop a contrastive learning approach that does not require identity annotations, allowing us to train the mapping network in a self-supervised manner.

**Constructing Triplets.** We prepare multiple combinations of prompts and latent codes. As illustrated in Figure 2, the same prompts (*e.g.* "a man on a soccer field ...") are paired with different latent codes, and a similar prompt (*e.g.* "an image captures a man playing soccer ...") is paired with the same latent code. The network is trained to generate pseudo-words that, when inserted into prompts, guide the synthesis of specific instances.

Specifically, we prepare an image and a triplet of prompts for each training sample. The triplet consists of (a) an anchor prompt, (b) a positive prompt, and (c) a negative prompt. The anchor prompt is a caption describing the image, modulated by a latent code. The positive prompt is another description of the image, modulated with the same latent code. The negative prompt has the same caption as the anchor prompt but is modulated with a different latent code. Formally, let $\boldsymbol{e}_1$ and $\boldsymbol{e}_2$ denote the embeddings of two descriptions $P_1$ and $P_2$ of the image $I$. Here, $\boldsymbol{z}_1$ and $\boldsymbol{z}_2$ denote two different latent codes, and $i$ and $j$ are the locations of target concept tokens in $\boldsymbol{e}_1$ and $\boldsymbol{e}_2$, respectively. We create an anchor prompt $\boldsymbol{e}^*$, a positive prompt $\boldsymbol{e}^+$,

and a negative prompt $\boldsymbol{e}^-$ via

$$
\begin{aligned}
\boldsymbol{e}^* &= \texttt{insert}(\boldsymbol{e}_1, \boldsymbol{w}_1, i), \quad \boldsymbol{e}^+ = \texttt{insert}(\boldsymbol{e}_2, \boldsymbol{w}_1, j), \qquad \boldsymbol{w}_1 = f(\boldsymbol{z}_1), \\
\boldsymbol{e}^- &= \texttt{insert}(\boldsymbol{e}_1, \boldsymbol{w}_2, i), \qquad\qquad\qquad\qquad\qquad\quad \boldsymbol{w}_2 = f(\boldsymbol{z}_2).
\end{aligned}
\tag{5}
$$

Then, we construct three noisy image latents to be paired with the three prompt embeddings. Since, in practice, a user may perform multiple generations with different initial noises, we obtain the noisy image latents by adding two different noises, $\epsilon_1$ and $\epsilon_2$, sampled from a normal distribution, to the image latent $\boldsymbol{x}$ to maintain subject consistency across multiple generations. We add the same noise to the anchor and negative samples to create a challenging scenario:

$$
\boldsymbol{x}_t^* = \sqrt{\alpha_t}\boldsymbol{x} + \sqrt{1-\alpha_t}\epsilon_1, \quad \boldsymbol{x}_t^+ = \sqrt{\alpha_t}\boldsymbol{x} + \sqrt{1-\alpha_t}\epsilon_2, \quad \boldsymbol{x}_t^- = \sqrt{\alpha_t}\boldsymbol{x} + \sqrt{1-\alpha_t}\epsilon_1,
\tag{6}
$$

where $\boldsymbol{x}_t^*$, $\boldsymbol{x}_t^+$, and $\boldsymbol{x}_t^-$ denote the anchor, positive, and negative noisy image latents, which are then paired with the modulated prompt embeddings $\boldsymbol{e}^*$, $\boldsymbol{e}^+$, and $\boldsymbol{e}^-$, respectively, to form the inputs to the denoiser.

**Building Association.** To encourage the generative model to synthesize subjects associated with pseudo-words, we apply a triplet loss (Schroff et al., 2015) to the denoiser outputs $\hat{\epsilon}^*$, $\hat{\epsilon}^+$, and $\hat{\epsilon}^-$ of the anchor, positive, and negative samples, pulling the anchor and positive samples closer while pushing the negative sample away. Since consistency is meaningful only in image latents rather than noise, we first obtain the predicted image latents $\hat{\boldsymbol{x}}^*$, $\hat{\boldsymbol{x}}^+$, and $\hat{\boldsymbol{x}}^-$ with DDIM (Song et al., 2021) approximation. Then, the distances between the three approximated latents are measured via

$$
\mathcal{L}_{\text{con}} = \mathcal{L}_{\text{dis}}(\hat{\boldsymbol{x}}^*, \hat{\boldsymbol{x}}^+) + \lambda_{\text{neg}} \cdot \frac{1}{\mathcal{L}_{\text{dis}}(\hat{\boldsymbol{x}}^*, \hat{\boldsymbol{x}}^-)},
\tag{7}
$$

where $\mathcal{L}_{\text{dis}}$ denotes a distance function. Note that we empirically find that the common form of triplet loss with subtraction leads to less distinction between different input latent codes; thus, our triplet loss is based on the reciprocal of the distance between the anchor and negative samples.

**Subject Masks.** Furthermore, since we only pursue subject consistency across images rather than similarity of entire images, we calculate the loss only in subject regions by applying masks to the output images. Subject masks can be obtained through an off-the-shelf referring segmentation model (Kirillov et al., 2023; Liu et al., 2023b; Ren et al., 2024), which annotates pixels corresponding to input words. Let $m$ denote the mask with boolean values that covers the pixels of target concepts. We replace $\mathcal{L}_{\text{dis}}(\cdot, \cdot)$ with $\mathcal{L}_{\text{dis}}^m(\cdot, \cdot)$ in Eq. (7) to denote the masked distance function with mask $m$, where $\mathcal{L}_{\text{dis}}^m(x, y) = \mathcal{L}_{\text{dis}}(m \cdot x, m \cdot y)$.

**Background Preservation.** With the aforementioned subject mask $m$, we negate the subject mask to obtain the background mask $\tilde{m} = 1 - m$. The background preservation loss is defined to minimize the distance between the backgrounds of the images generated with and without the pseudo-words:

$$
\mathcal{L}_{\text{back}} = \mathcal{L}_{\text{dis}}^{\tilde{m}}(\hat{\boldsymbol{x}}^*, \hat{\boldsymbol{x}}_1) + \mathcal{L}_{\text{dis}}^{\tilde{m}}(\hat{\boldsymbol{x}}^+, \hat{\boldsymbol{x}}_2) + \mathcal{L}_{\text{dis}}^{\tilde{m}}(\hat{\boldsymbol{x}}^-, \hat{\boldsymbol{x}}_1).
\tag{8}
$$

Here $\hat{\boldsymbol{x}}_1$ and $\hat{\boldsymbol{x}}_2$ are DDIM approximations of the denoiser outputs $\epsilon_\theta(\boldsymbol{x}_t, \boldsymbol{e}_1, t)$ and $\epsilon_\theta(\boldsymbol{x}_t, \boldsymbol{e}_2, t)$, respectively. The final loss function consists of the contrastive and background preservation losses:

$$
\mathcal{L} = \lambda_{\text{con}} \cdot \mathcal{L}_{\text{con}} + \lambda_{\text{back}} \cdot \mathcal{L}_{\text{back}},
\tag{9}
$$

where $\lambda_{\text{con}}$ and $\lambda_{\text{back}}$ are weights balancing the two losses.

## 4 Experimental Results

### 4.1 Implementation Details

**Architecture.** We implement the mapping network $f$ as an $n$-layer MLP and leverage Stable Diffusion XL (Podell et al., 2023) as the text-to-image diffusion model. We train only the mapping network $f$, with all other model weights frozen.

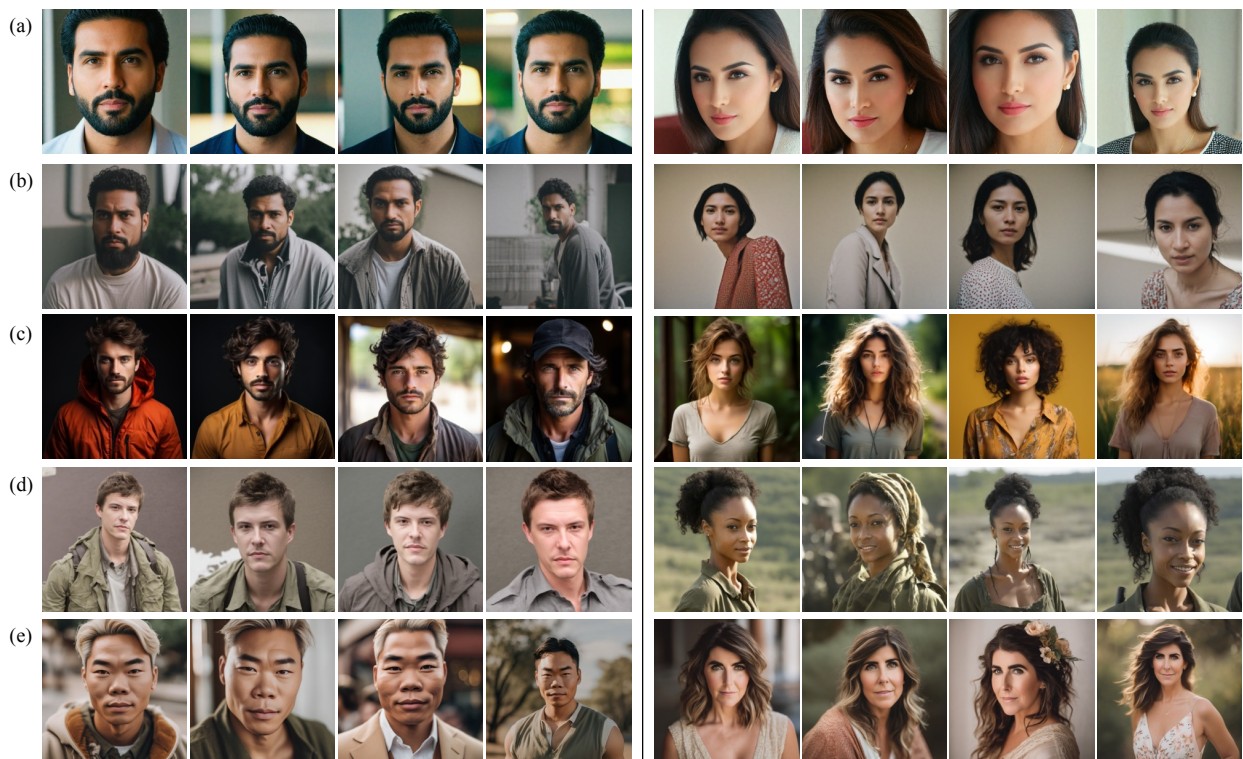

Figure 3. **Qualitative comparisons on *Portraits*** from (a) StoryDiffusion (Zhou et al., 2024), (b) Consistory (Tewel et al., 2024), (c) 1Prompt1Story (Liu et al., 2025), (d) DreamBooth (Ruiz et al., 2022), and (e) CoCoIns. The left four columns are generated with a man as the subject, and the right four with a woman. We achieve subject consistency without generating images in batches or performing reference fine-tuning.

**Negative Distance Weight Schedule.** We empirically find that the distance between anchor and negative samples is often too large at the beginning of training. The model tends to ignore the randomly initialized input latent codes and produces identical outputs. Therefore, we implement the weight of negative distance $\lambda_{\text{neg}} = \gamma(k/K)^\beta$ as an increasing function over training steps, where $k$ and $K$ are the current and total training steps, and $\gamma$ and $\beta$ are hyperparameters.

## 4.2 Experiment Setups

We conduct comprehensive experiments on single-subject human faces with our approach, followed by multiple subjects and other object categories. More details on data collection can be found in Appendix D.

**Training.** We train the mapping network using the CelebA dataset (Liu et al., 2015), which comprises 20K images and 10K identities. We generate prompts with the captioning model LLaVA-Next (Liu et al., 2023a) and masks with the zero-shot referring segmentation model Grounded SAM 2 (Kirillov et al., 2023; Liu et al., 2023b; Ren et al., 2024).

**Evaluation.** We design two prompt sets for comprehensive evaluation:

- *Portraits*: This set evaluates face similarity with clear frontal faces. It contains 1K sentences composed of the template "A `[subject]` is looking at the camera.", where `[subject]` is one of {`man`, `woman`, `boy`, `girl`, `person`}. Each subject contains 200 sentences, resulting in a total of 1K samples.
- *Scenes*: This set represents real-world performance with free-form prompts, where face poses and angles vary. It comprises 1K sentences generated by a Large Language Model (LLM) (Ouyang et al., 2022). We prompt the LLM to generate sentences of the same five subjects doing something in diverse situations, including four settings: daily lives, professional environments, cultural or recreational occasions, and outdoor activities, each with 50 samples.

Table 1. **Quantitative performance on *Portraits*.** We achieve comparable consistency and better diversity compared to approaches that generate images in batches.

|  | Sim↑ | Div↑ | CLIP↑ |
|---|---|---|---|
| CelebA | 0.590 | 0.992 | 0.299 |
| Consistory | 0.356 | 0.774 | 0.218 |
| StoryDiffusion | **0.637** | 0.577 | 0.217 |
| 1Prompt1Story | 0.307 | 0.611 | **0.228** |
| Ours | 0.600 | **0.799** | 0.193 |

Table 2. **Quantitative performance on *Scenes*.** We achieve the best face similarity while maintaining similar subject diversity and prompt fidelity compared to other methods. In addition, we generate images independently, enabling high flexibility for future generations.

|  | Sim↑ | Div↑ | CLIP↑ | DS↑ |
|---|---|---|---|---|
| Consistory | 0.098 | **0.883** | **0.297** | 0.383 |
| StoryDiffusion | 0.159 | 0.814 | 0.290 | **0.407** |
| Ours | **0.256** | 0.847 | 0.290 | 0.388 |

We measure subject similarity and diversity over *Portraits* and *Scenes*. For faces, we estimate face similarity (Sim) and diversity (Div) of cropped and aligned images. Face similarity is the pairwise cosine similarity of ArcFace (Deng et al., 2022) embeddings between images with the same identity. To estimate diversity, we first average the face embeddings for each identity. We then calculate the pairwise cosine similarity between the averaged embeddings of all identities.

Since *Scenes* considers images in real-world applications where faces are not always clear and large, we calculate DreamSim (Fu et al., 2023) (DS), a learned perceptual distance aligned with human preference, for subject similarity. We also measure prompt fidelity for both sets using CLIP, which computes the cosine similarity between the projected embeddings of the CLIP text and image encoders.

**Evaluated Methods.** We evaluate our method against tuning-free subject-consistent generation and tuning-based customization. Tuning-free schemes include Consistory (Tewel et al., 2024), StoryDiffusion (Zhou et al., 2024), and 1Prompt1Story (Liu et al., 2025). Since tuning-based methods require fine-tuning for all subjects, which incurs heavy computational costs, we use DreamBooth (Ruiz et al., 2022) as an example and present only quantitative results.

### 4.3 Empirical Results

Table 1 shows the quantitative performance on *Portraits*, and Figure 3 displays two subjects, a man and a woman, each with four images, generated by all approaches. We measure the similarity (Sim) and diversity (Div) of the training dataset CelebA for reference. Although StoryDiffusion exhibits the highest similarity, even surpassing CelebA, its diversity remains low. Subjects generated from different batches with different initial noise converge toward a similar appearance. Our approach achieves comparable similarity to StoryDiffusion and CelebA while generating diverse subjects.

We demonstrate examples from *Scenes* in Figure 4 and present the quantitative performance in Table 2. We compare our approach against Consistory and StoryDiffusion because 1Prompt1Story does not support free-form prompts. It operates with a specific prompt structure where a subject description is followed by multiple context descriptions. Our model performs favorably on face similarity while performing comparably on diversity and fidelity.

### 4.4 Ablation Study

**Consistency Loss.** We analyze the efficacy of triplet loss in maintaining consistency. We calculate face similarity and diversity on *Portraits* and evaluate prompt fidelity on *Scenes*. Table 3 shows the performance of applying the loss of only positive distance (Pos), positive and negative distance (Pos + Neg), using the common form of triplet loss with negative distance subtraction (subtract), and setting the weight of negative distance as an increasing schedule over training iterations. The four settings yield similar prompt fidelity, while implementing negative distance, the reciprocal form, and the weighting schedule enhances performance.

**Background Loss.** We also evaluate the effect of removing background preservation loss under the same test sets. As shown in Table 4, using only consistency loss results in unsatisfactory face similarity and fidelity.

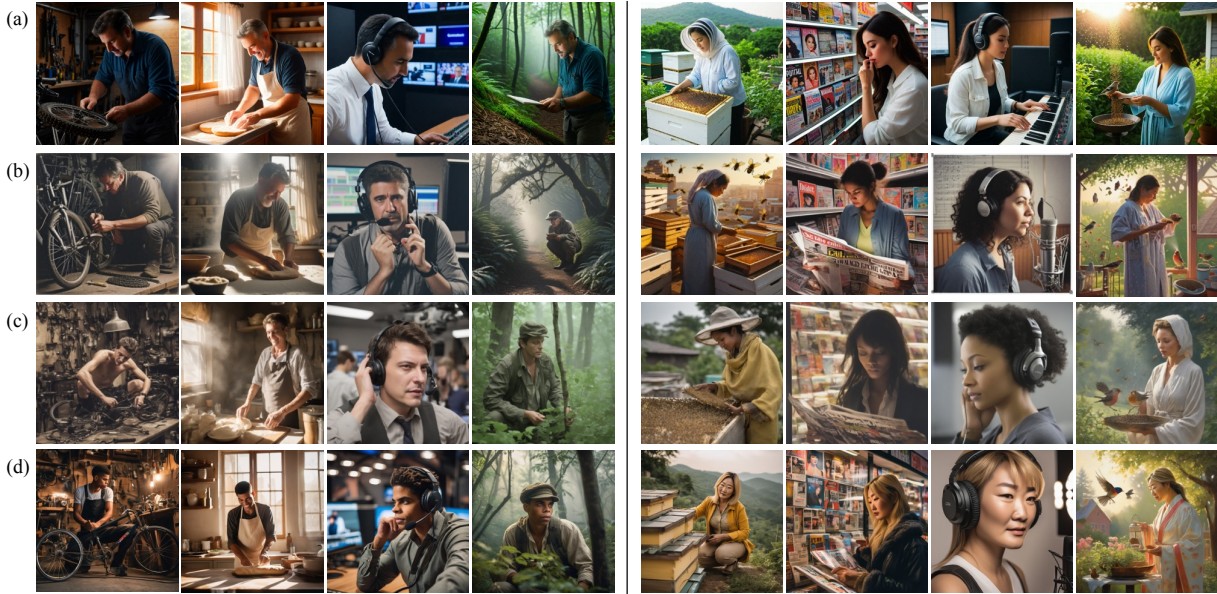

Figure 4. **Qualitative comparisons on *Scenes*** from (a) StoryDiffusion Zhou et al. (2024), (b) Consistory Tewel et al. (2024), (c) DreamBooth Ruiz et al. (2022), and (d) CoCoIns. 1Prompt1Story Liu et al. (2025) is absent because it requires a specific prompt format with unified subject descriptions. The left and right four columns show two different subjects in diverse contexts. The prompts can be found in Appendix D.

Table 3. **Performance of ablating consistency loss.** We train the network with the distance from the positive sample (Pos) and the reciprocal of the negative sample (Neg). Instead of the common triplet loss with subtraction of negative distances (subtract), we minimize its reciprocal and apply a weighting schedule that increases over training iterations (Schedule). The final setting achieves the best face similarity and diversity with similar prompt fidelity.

|  | Sim↑ | Div↑ | CLIP↑ |
|---|---|---|---|
| Pos | 0.394 | 0.500 | 0.293 |
| Pos + Neg | 0.492 | 0.750 | 0.290 |
| Pos + Neg (subtract) | 0.380 | 0.444 | **0.294** |
| Pos + Neg + Schedule | **0.600** | **0.799** | 0.290 |

Table 4. **Performance of ablating background loss**. In addition to the triplet loss for consistency (Consistency Loss), we adopt a segmentation mask (Mask) to control the variation region and a background preservation loss (Background) to keep backgrounds close to the original predictions, thereby significantly improving face diversity and prompt fidelity while maintaining similar face similarity.

|  | Sim↑ | Div↑ | CLIP↑ |
|---|---|---|---|
| Consistency Loss | 0.444 | 0.395 | 0.138 |
| + Mask | **0.619** | 0.352 | 0.128 |
| + Mask + Background | 0.600 | **0.799** | **0.290** |

While training with masked distances (+ Mask) without preserving backgrounds yields slightly higher face similarity, it results in low diversity and fidelity. In this setting, although the loss is calculated only within masked regions, the model modifies non-masked regions, possibly due to the self-attention mechanism, which allows information to interact globally. Applying background preservation (+ Background) significantly improves diversity and fidelity.

**Prompt and Noise Combinations.** We compare strategies for constructing training triplets. We evaluate the performance of using the same two prompts or creating noisy latent images with the same noise for anchor and positive samples. Table 5 shows that using different prompts and noise achieves the best similarity and nearly the same prompt fidelity.

**Training with Subject Annotations.** We show results from directly training the mapping network as a noise prediction problem on CelebA. Figure 5 shows that the model learns to maintain subject consistency, but the output quality is also affected by the limited, low-quality data.

Table 5. **Performance of prompt and noise combinations for constructing training triplets.** Compared to using the same prompts (=) or noise for the anchor and positive samples, using two different (≠) prompts and noise yields the best face similarity and diversity with similar prompt fidelity.

| Prompts | Noise | Sim↑ | Div↑ | CLIP↑ |
|---------|-------|------|------|-------|
| = | ≠ | 0.548 | 0.686 | 0.290 |
| ≠ | = | 0.306 | 0.772 | **0.292** |
| ≠ | ≠ | **0.600** | **0.799** | 0.290 |

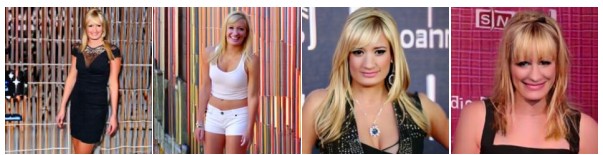

Figure 5. **Results of training the mapping network as a noise prediction problem.** The network overfits to the dataset and generates low-quality images.

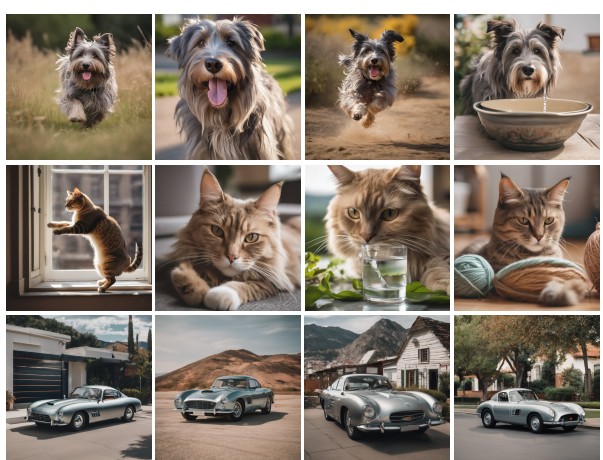

Figure 6. **Results of general concept consistency.** Our approach makes no assumptions about object categories. It can potentially be applied to other concepts such as cats, dogs, and cars.

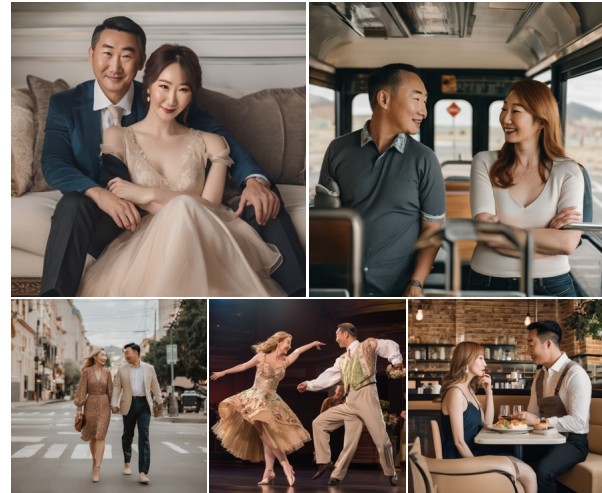

Figure 7. **Results of multi-subject consistency.** Given two different latent codes, the model trained on single-subject images can maintain consistency for multiple subjects.

### 4.5 Extensions

**General Concepts.** Since our approach imposes no constraints on subject classes, we also demonstrate that it can be applied to general concepts. We train the model with animal (Choi et al., 2020) and car (Yu et al., 2015) images. The examples in Figure 6 show that the model can potentially be applied to more concept categories beyond humans.

**Multi-Subject Consistency.** In addition to analyzing single subjects, we demonstrate our ability to maintain consistency across images with multiple subjects. Figure 7 contains two sets of examples of a man and a woman in different settings. We use two different input codes for the two subjects and generate the images with the model trained on single-subject data. While the model has never seen two faces in an image, it can identify face regions and maintain consistency for multiple subjects. Despite some entanglement and influence between subjects, the results demonstrate the potential to extend the model to multi-subject scenarios.

## 5 Conclusion

In this work, we propose Contrastive Concept Instantiation (CoCoIns), the first approach to achieve subject consistency without the need for time-consuming fine-tuning and labor-intensive reference collection. Our key idea is to model concept instances in a latent space and train a mapping network using contrastive learning to associate latent codes with output concept instances. We demonstrate its efficacy on single-subject human faces and extend it to multi-subject scenarios and general concepts. We believe this work establishes a foundation for controllable content creation.

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

## A   Comparison of Settings and Implementation

We provide a comparison of settings and implementation between our approach and prior work in Table 6. As discussed in Section 2, prior approaches can be categorized into subject-driven generation and subject-consistent generation.

Subject-driven approaches personalize a generative model by fine-tuning parameters on reference images or incorporating additional pretrained encoders. These approaches are either time-consuming due to subject-specific fine-tuning or require integrating general encoders, such as DINO, or domain-specific encoders, such as face recognition models.

Subject-consistent approaches modify the prompt or attention mechanisms of the base generator, thereby avoiding the need for extra modules, reference images, or additional training. Their main limitations are the requirement to generate images in batches or the reliance on stored features.

Our method is lighter and more flexible. It requires only a single MLP training and supports independent inference.

Table 6. Comparison of settings and implementation between our approach and prior work.

| Approach | Extra Modules | Reference | Training | Inference Constraints |
|---|---|---|---|---|
| Tuning-based Personalization | None | Yes | Subject Tuning | None |
| Encoder-based Personalization | Pretrained Encoders | Yes | Once | None |
| Subject-Consistent Generation | None | No | None | Batch |
| CoCoIns (Ours) | Lightweight MLP | No | Once | None |

## B   Additional Implementation Details

**Computational Resources.** The experiments are conducted on an AMD EPYC 9354 CPU and four NVIDIA A6000 GPUs. Each training round takes approximately eight hours.

**Hyperparameters.** We list the hyperparameters in Table 7. They are determined by grid search. The distance function $\mathcal{L}_{\text{dis}}$ is the Mean Squared Error.

**Latent Code Sampling.** Latent codes are randomly sampled from the Gaussian distribution $\mathcal{N}(\mathbf{0}, \boldsymbol{I})$, as stated in Eq. (3). In each training triplet, the anchor and positive samples share the same latent code, and the negative sample is paired with another randomly sampled code, as detailed in Eq. (5).

Table 7. Hyperparameters.

| Hyperparameters | Value |
|---|---|
| $c$ | 256 |
| $\lambda_{\text{con}}$ | 1 |
| $\lambda_{\text{back}}$ | 30 |
| $\gamma$ | 0.00001 |
| $\beta$ | 2 |
| $n$ | 8 |
| $K$ | 5000 |
| Batch Size | 128 |
| Learning Rate | 0.0001 |
| Learning Rate Decay | Cosine |
| Learning Rate Warmup | 500 |
| Optimizer | Adam |
| Weight Decay | 0.2 |

Table 8. FID score between each approach and its base models.

|  | FID ↓ |
| --- | --- |
| Consistory | 13.3 |
| StoryDiffusion | 15.2 |
| Ours | 9.2 |

Table 9. Comparison of performance using MSE and DreamSim loss as the distance function.

| Distance | Sim ↑ | Div ↑ | CLIP ↑ |
| --- | --- | --- | --- |
| MSE | 0.600 | 0.799 | 0.193 |
| DreamSim | 0.314 | 0.724 | 0.291 |

Table 10. Ablation study on weighting background loss.

| $\lambda_{back}$ | Sim ↑ | Div ↑ | CLIP ↑ |
| --- | --- | --- | --- |
| 10 | 0.637 | 0.603 | 0.265 |
| 30 | 0.600 | 0.799 | 0.290 |
| 50 | 0.516 | 0.707 | 0.292 |

Table 11. Ablation study on weighting negative distances.

| $\gamma$ | Sim ↑ | Div ↑ | CLIP ↑ |
| --- | --- | --- | --- |
| $10^{-4}$ | 0.666 | 0.766 | 0.288 |
| $10^{-5}$ | 0.600 | 0.799 | 0.290 |
| $10^{-6}$ | 0.445 | 0.635 | 0.293 |

Table 12. Ablation study on schedules of negative distance weighting.

| $\beta$ | Sim ↑ | Div ↑ | CLIP ↑ |
| --- | --- | --- | --- |
| 1 | 0.528 | 0.735 | 0.291 |
| 2 | 0.600 | 0.799 | 0.290 |
| 3 | 0.520 | 0.710 | 0.291 |

## C  Additional Experimental Results

### C.1  Additional Comparison

**Image Quality.** We evaluate the quality of the generated images using the Fréchet Inception Distance (FID) score. Since various approaches are based on different models, we compute FID scores by comparing the images produced by each approach with those generated by its respective base model. Consistory and our method use SDXL; StoryDiffusion utilizes RealVisXL. Given that the FID score is sensitive to sample size, we duplicate the *Scenes* dataset ten times to create a total of 10K prompts. We then generate 10K images using both the compared approaches and their base models. As shown in Table 8, our generated images are more closely aligned with the distribution of the base model.

**Distance Function.** We investigate the effect of different distance functions for measuring subject similarity during training. In our training strategy, the model is trained to generate similar appearances from the same image corrupted by two different noises, using two similar prompts and the same latent code. When reconstruction is nearly perfect, a simple pixel-wise metric such as Mean Squared Error (MSE) is sufficient. However, in more realistic scenarios where outputs are imperfect, we may need a subject-aware similarity measure. One option is to use a subject encoder; however, existing encoders are typically designed for narrow domains (*e.g.* face recognition) and often rely on non-differentiable operations, such as cropping and landmark alignment, making them unsuitable for end-to-end training. An alternative is a learned perceptual loss, such as DreamSim (Fu et al., 2023), which is trained to align with human perception of similarity. However, its notion of similarity may reflect factors like layout and color rather than subject identity.

Therefore, we primarily use MSE as our distance function but also include a comparison with DreamSim. To apply DreamSim as a perceptual similarity metric, we decode the DDIM-approximated image latents using the autoencoder and compute the cosine similarity between their DreamSim embeddings. We use the DINOv2 checkpoint for DreamSim. As shown in Table 9, compared to MSE, DreamSim loss struggles to generate consistent subjects.

### C.2  Additional Ablation Study

We examine the impact of hyperparameter choices on performance, specifically (1) the balance between the consistency and background loss and (2) the weighting schedule of negative distances.

Table 10 shows the comparison between different background loss weights, denoted by $\lambda_{back}$ in Eq. (9). Increasing the importance of background loss improves prompt fidelity at the cost of lower face similarity and diversity. We choose the $\lambda_{back}$ that balances these factors.

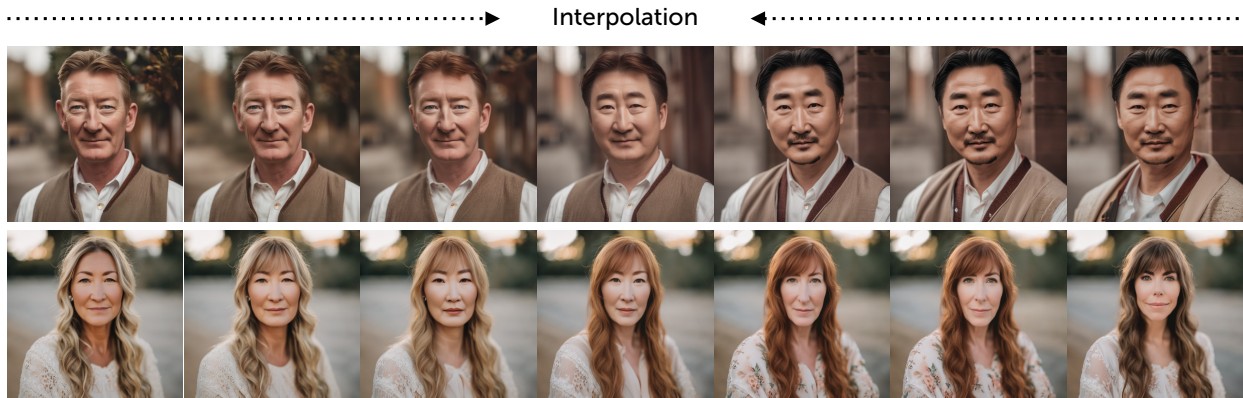

Figure 8. **Results generated with interpolations of two latent codes.** The leftmost and rightmost images are generated from two randomly sampled codes. The intermediate images are the results of interpolations between the two codes, demonstrating the gradual transition between the two faces.

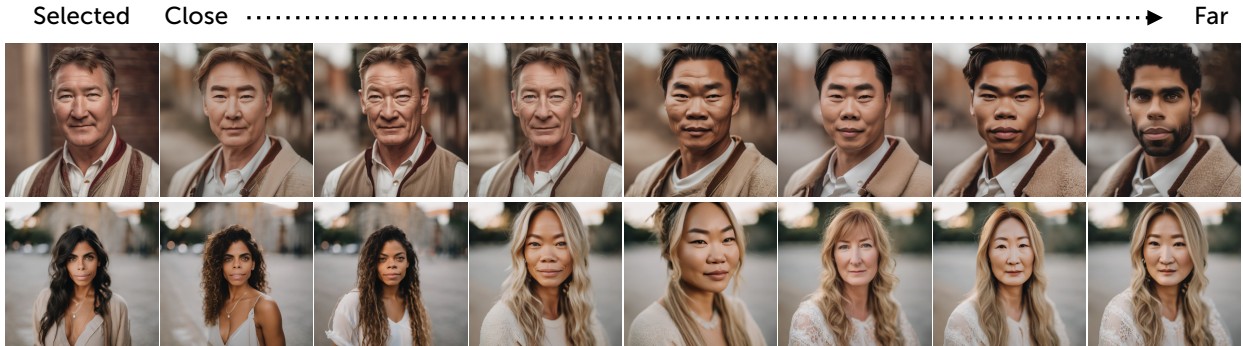

Figure 9. **Results generated with neighbors of a pseudo-word.** We generate 100 images with randomly sampled latent codes and a fixed initial noise. Given a randomly selected code and its corresponding image, we find the neighbors of the selected code by sorting the cosine similarity between the pseudo-word transformed from the code and the others. The result shows that the pseudo-words of closer neighbors (*i.e.* high cosine similarity) produce more similar faces.

Additionally, as discussed in Section 4.1, the weighting of negative distances is implemented as an increasing schedule parameterized by $\gamma(k/K)^{\beta}$, where $k$ is the current training step, and $K$ is the total number of training steps. $\gamma$ and $\beta$ are hyperparameters. We also examine the effect of varying these two hyperparameters.

Table 11 presents the results for different values of $\gamma$, and Table 12 provides comparisons between different values of $\beta$. Similar to our previous study on consistency and background loss, some hyperparameter sets lead to higher face similarity but lower prompt fidelity or diversity. We choose the hyperparameter set that balances all of these important metrics.

### C.3 Analysis of the Latent Space

We analyze the latent space that models the instance distribution of concepts. To better understand the relationships between latent codes in the space and the information captured by pseudo-words, we demonstrate results generated with interpolations of latent codes and neighbors of pseudo-words.

**Interpolation of Latent Codes.** We reveal the relationships between latent codes by visualizing their interpolations. Given two randomly sampled latent codes, we generate images by gradually interpolating between them, starting from a fixed initial noise. As illustrated in the two examples in Figure 8, the leftmost and rightmost images correspond to the two original latent codes, while the intermediate images depict the

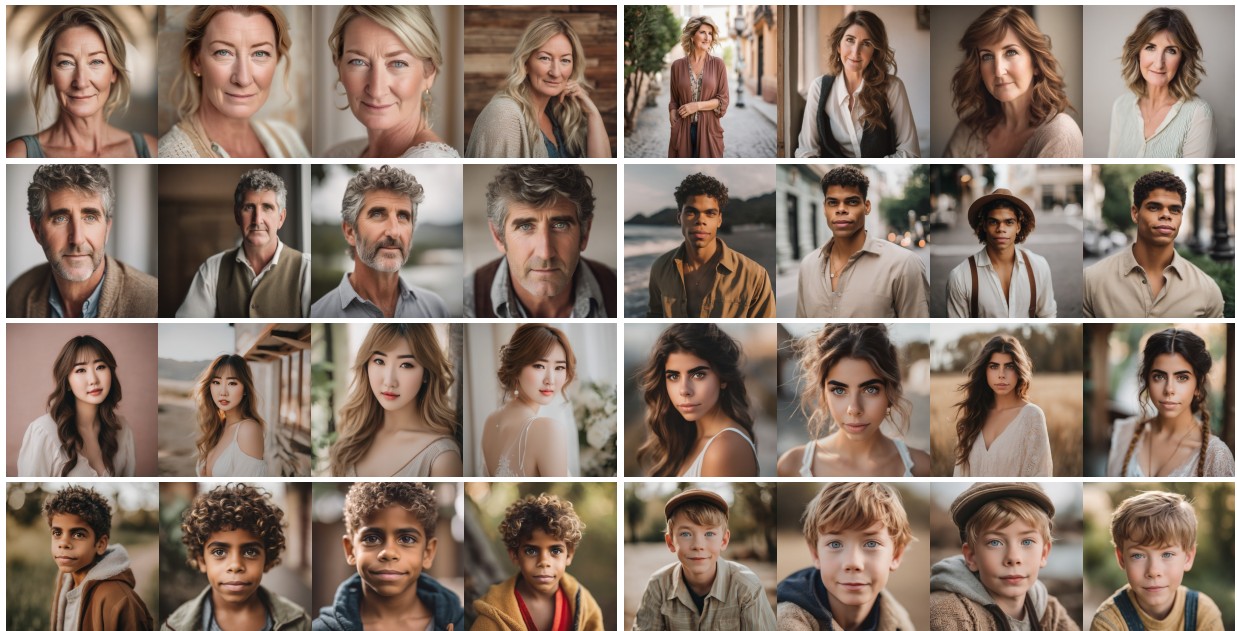

Figure 10. **Eight sets of examples from *Portraits*.** The subjects for each row are woman, man, girl, and boy.

transition between them. Using a fixed initial noise results in similar background appearances across the interpolated images.

**Neighbors of Pseudo-Words.** We also illustrate the information captured by a pseudo-word by retrieving its neighbors. Specifically, we randomly sample 100 latent codes and generate images from them. We then select one code and its corresponding image, and compute the cosine similarity between the pseudo-word derived from this code and those from the remaining codes. The other images are sorted based on this similarity. As shown in Figure 9, the leftmost image is generated from the selected latent code, while the images to its right are arranged in descending order of pseudo-word similarity. The results indicate that closer neighbors, *i.e.* pseudo-words with higher similarity, tend to generate more visually similar faces.

### C.4 Additional Generated Samples

We provide additional samples from *Portraits* in Figure 10 and *Scenes* in Figure 11. Each row contains two subjects with four examples. The subjects for each row are woman, man, girl, and boy.

More multi-subject examples are in Figure 12. We also include comparisons with two previous approaches, Consistory and StoryDiffusion. Our examples demonstrate high image quality and consistency. The prompts in Figure 7 and Figure 12 are "a man and a woman sitting on the sofa", "a man and a woman taking the bus", "a man and a woman walking on the street", and "a man and a woman dancing on the stage".

More examples in Figure 13 demonstrate consistency for general concepts. The prompts in Figure 6 and Figure 13 are "a photo of a cat", "dog", or "car".

## D    Data Collection and Generation

**Identifying Subjects in Descriptions.** We generate descriptions for training images with a multi-modal captioner (Liu et al., 2023a). Then, we prompt an LLM to identify the words that are most likely to be the subjects in the sentences. The prompt is as follows:

> These are two captions of an image. Tag the words related to the subjects of the captions.
> Return the captions in a JSON with keys "caption1" and "caption2".

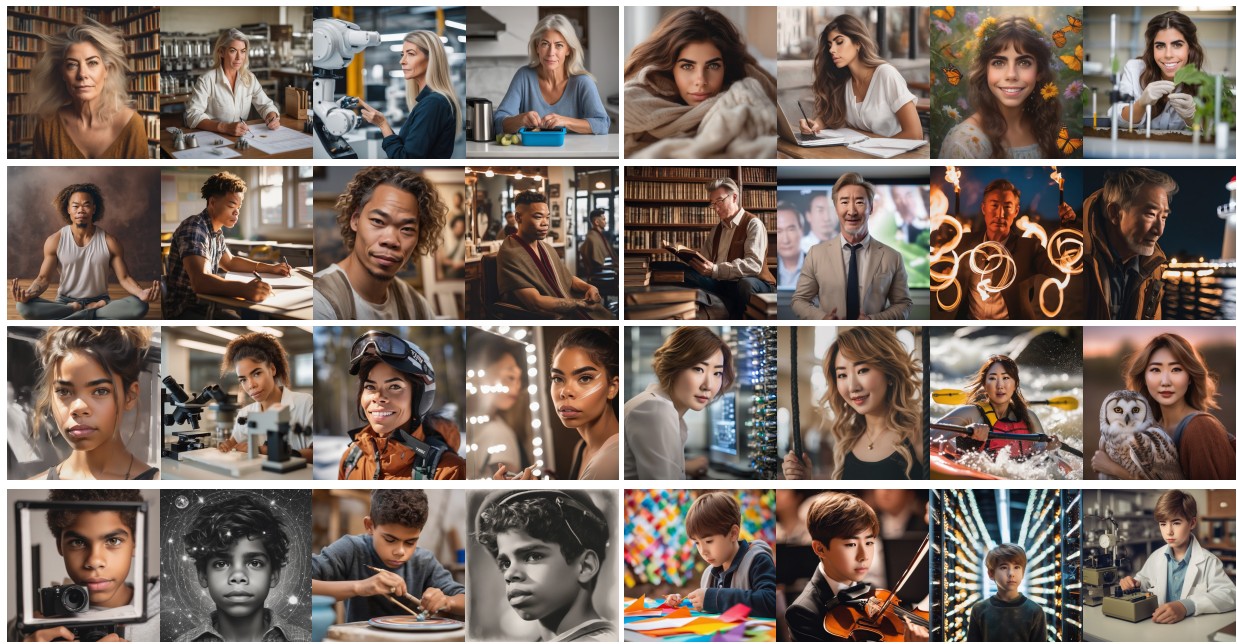

Figure 11. **Eight sets of examples from *Scenes*.** The subjects for each row are woman, man, girl, and boy.

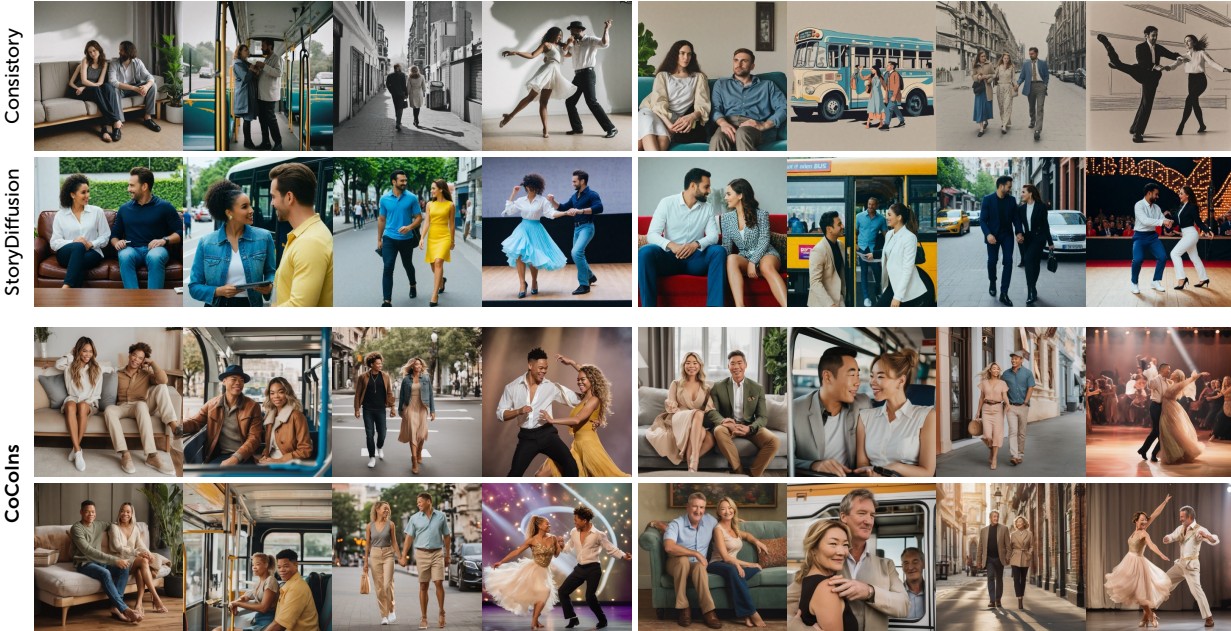

Figure 12. More examples and comparison with previous work of multi-subject consistency.

1. Identify words related to a person: Look for specific nouns or noun phrases that refer to a person. Exclude pronouns (e.g., "he", "she", "they", "it") and collective nouns (e.g., "people") from tagging.

2. Tag these words: Surround each identified word or noun phrase with `<subj>` and `</subj>` tags. Use `<subj1>`, `<subj2>`, etc., for different items. Apply the same index number to all references to the same item.

3. If no relevant words are found: Return the original captions without any changes.

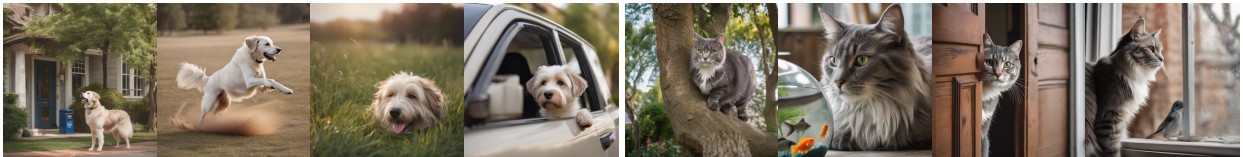

Figure 13. Two sets of examples demonstrating consistency for general concepts.

4. Handling ambiguity: If a word has ambiguous references or unclear roles, do not tag it.

5. Pronouns: Do not tag pronouns.

Then, we implement a tag parser to provide subject locations during training. In inference, users can provide locations to indicate target concepts that must be consistent.

**Generating Diverse Scenes for Evaluation.** One of the test datasets is collected to represent diverse, real-world contexts. Taking "man" as an example, we prompt an LLM to generate these sentences using the following instruction:

Generate fifty scene descriptions of a man in daily life.

1. Focus on the subject

- The subject should always be a "man".
- Provide descriptions of diverse scenes that feature a specific subject performing an action in a certain place.

2. Make actions and locations diverse

- Always use a verb and a location that has not appeared in previous sentences.
- The scenes should be related to everyday life, such as cooking, driving, walking, reading, etc.

3. Portrait Details:

- The descriptions should feature close-up views of the subject's face.
- Sensory details should enhance the scene (lighting, surroundings, sounds, smells, etc.), but keep the focus on the subject's face. The environment should be vivid but relevant to the subject's action or setting.

4. Tag the subjects:

- Tag the subject with `<subj1> </subj1>`. For example,"a `<subj1>man</subj1>` stands in the room"
- Do not tag pronouns (such as "he", "his", "him", etc.).
- If there are multiple subjects, use different indexes for each individual (e.g., `<subj1>`, `<subj2>`, etc.).
- Use the same index for all references to the same subject.

5. Length: No more than three sentences.

Now generate 50 more samples with scenes related to technical or professional settings. For example, locations can be offices, schools, hospitals, labs, farms, factories, studios, kitchens, etc.

Now generate 50 more samples with scenes related to casual, cultural, or recreational occasions, such as dances, music, dramas, movies, sports, arts, etc.

Now generate 50 more samples with scenes related to outdoor activities or in nature, such as gardens, parks, mountains, forests, beaches, rivers, lakes, etc.

**Prompts of Qualitative Comparison.** The images in Figure 4 are generated with the following prompts:

- A man fixes his bicycle chain in the workshop, grease streaking his concentrated face. The smell of rubber and oil surrounds him as he works by lamplight.
- A man kneads bread dough in his sun-drenched kitchen, flour dusting his smile lines as morning light streams through gauzy curtains. His focused gaze follows the rhythmic movements of his hands, while the yeasty aroma fills the warm air.
- At the television studio, a man directs a live broadcast, speaking calmly into his headset. His focused gaze darts between multiple monitors as he calls camera changes.
- A man tracks wildlife in the misty forest, his experienced eyes reading subtle signs in the undergrowth. The early morning light filters through the canopy, illuminating his weathered face as he studies fresh prints.
- A woman tends to her rooftop beehives, moving with calm confidence among the buzzing insects. Her peaceful expression reflects years of experience as she checks each frame.
- Under fluorescent lights at the corner store, a woman browses magazine covers, her reflection ghosted in the glossy pages. Her fingers trace headlines as she squints slightly, the artificial brightness highlighting the fine lines around her eyes.
- In the acoustically treated recording studio, a woman masters audio tracks, her trained ear catching subtle nuances. Her eyes close briefly as she adjusts levels with precise movements.
- A woman fills her bird feeder in the backyard, morning dew soaking the hem of her robe. She squints against the rising sun, watching finches dart around her head as she pours the seeds.

**Tackling Subject Ambiguity.** Our experiments focus on single-subject consistency. To create a clean training dataset and minimize ambiguity, we use CelebA (Liu et al., 2015), which primarily contains human portraits. We generate masks using a powerful off-the-shelf model, Grounded SAM 2 (Ren et al., 2024). We then manually filter out images with more than one face according to the segmentation results. This step ensures that the training images contain single subjects and reduces ambiguous masks.

**Extension to Multi-Subject Scenes.** Although the model is trained only on single-subject portrait images, it can handle subject consistency in more challenging conditions, such as free-form prompts in *Scenes* and images with two subjects. For more challenging scenarios with multi-subject images, the proposed loss function framework can be further extended. Since prior work (Hertz et al., 2022) has shown the relationships between word embeddings and feature maps, and a pseudo-word operates in the text embedding space to describe the subject that follows, we hypothesize that the model can learn to associate a pseudo-word with features of its corresponding subject, even in a multi-subject scene.

More thorough experimentation and evaluation are needed for multi-subject scenarios, which will be part of our future work.

**Segmentation Accuracy.** In single-subject experiments, inaccurate segmentation is less pronounced. Segmenting a person in a portrait, as is common in the CelebA dataset, is relatively straightforward for a powerful model like Grounded SAM 2. In addition, we manually validate 100 randomly sampled images and find that the predictions are consistently reliable for this use case.

**Licenses.** The main experiments are conducted on CelebA (Liu et al., 2015). It is made available for noncommercial research purposes and requires users to comply with the terms outlined in the official usage agreement.

# E  Further Discussions

**Preventing Prompt Corruption.** To ensure that a pseudo-word affects only the subject and does not corrupt the rest of the prompt, we use a background-preservation loss. This loss minimizes the difference in the background regions between an image generated with the pseudo-word and one generated without it. This localizes the effect of a pseudo-word to the masked subject region, thereby preserving the overall scene context dictated by the original prompt. The ablation study in Table 4 shows that removing this loss significantly harms prompt fidelity.

**Place of Pseudo-Words.** CoCoIns is developed based on the assumption that the pseudo-word lies in the text embedding space. Our mapping network $f$ is explicitly designed to take a latent code $z$ and output

a pseudo-word $w$ as a vector that has the same dimension as the text embeddings of the pretrained text encoder. This pseudo-word vector is then inserted directly into the sequence of prompt embeddings before the subject token.

**Aligning Pseudo-Words.** The consistency loss is not defined directly on the pseudo-words. Instead, a pseudo-word is made meaningful through an indirect alignment process via a contrastive loss applied to the generated images.

We use a triplet loss applied to the predicted image latents from the denoiser. The model is trained to minimize the visual difference between images generated with the same pseudo-word (even with different prompts and noise) while maximizing the visual difference between images generated with different pseudo-words. This process forces the model to treat each pseudo-word as a unique identifier for a specific visual appearance. The meaning of a pseudo-word becomes the consistent subject identity it produces.

We also analyze the space of pseudo-words in Appendix C.3, which provides strong evidence that this indirect alignment is successful. The experimental results show that:

- **Interpolation**: Smoothly interpolating between two latent codes results in a smooth visual transition between the two corresponding faces.
- **Similarity**: Pseudo-words that are closer to each other in the embedding space (*i.e.* have higher cosine similarity) produce subjects with more similar appearances.

These results demonstrate that the learned space of pseudo-words is well-structured and meaningful.

## F  Broader Impacts

Our approach enables image generative models to maintain subject consistency, extending their applicability across various tasks and modalities. Although this versatility could potentially be misused, such risks can be effectively mitigated through responsible deployment strategies, including strict usage policies, gated releases, and watermarking techniques.

