# OpenReview forum: "CoCoIns: Consistent Subject Generation via Contrastive Instantiated Concepts"
_TMLR — Accepted by TMLR_

### Review · Reviewer_3wKr · 2025-07-22

**Summary Of Contributions:**

This paper mainly targets subject-driven generations. The authors point out that previous methods mainly rely on time-consuming tuning or referencs for all subjects, leading to limited usage. To this end they propose a new method which maps latent codes to pseudo word embeddings, which are merged into text prompt and used in the imgae generation process. The authors leverage contrastive learning to train this framework. They conduct several experiments to validate the effectiveness of the proposed method.

**Audience:**

No

**Audience Explanation:**

While this paper is indeed handling a widely studied problem, i.e. the subject-consistent generation, the specific setting is essentially different from the previous works. The reason that this task is important is that one can generate consistent images based on some reference images. However, the proposed method can only control the ID information by latent code, but not some extra source knowledge. This makes the method less useful compared to previous ones.

**Claims And Evidence:**

No

**Claims Explanation:**

1. For quantitative experiment, the proposed mehod fails to perform the best in terms of almost all of the metrics.
2. For qualitative experiment, it is obvious that the generated images by the proposed method as shown in Fig.3(e) and Fig.7 has severe deformed human faces. This leads to the concern that the proposed method can damage the generation ability of the pretrained model.
3. Given that more advanced diffusion-based models are commonly used in recent papers, it is insufficient to only leverage SDXL in the experiments.
4. There is no specific metric or visualization to show the effectiveness of contrastive learning.
5. One can find that the dogs and cars in Fig.6 are not consistent among the images.

**Requested Changes:**

Please refer to the above comments.

---

> ### Author Response · Authors · 2025-08-15
> **Author Response (1/2)**
>
> We appreciate the valuable review and address the raised concerns as follows.
>
> ## 1 Quantitative Evaluation
>
> We quantitatively evaluate our method against batch-generated approaches, where samples within the same batch can access other samples. This represents a constrained setting that is not always practical. In contrast, our method addresses a flexible scenario where users **generate images individually**, achieving performance comparable to these batch-based approaches.
>
> Table 1 in the main paper provides face similarity, diversity, and prompt fidelity metrics estimated from CelebA, a real-world dataset containing front-face images in diverse scenes. Compared to CelebA, our method achieves **similar face similarity** on the test set _Portraits_, which focuses on front-face images, and produces **similar prompt fidelity** on the test set _Scenes_, which contains prompts describing diverse environments.
>
> We also observe that the three metrics — similarity, diversity, and prompt fidelity — **complement each other and should be considered jointly**. For example, StoryDiffusion achieves high face similarity on _Portraits_ but at the cost of low diversity, while Consistory obtains high diversity and fidelity on _Scenes_ but struggles to maintain consistency. In contrast, our method balances all three metrics and delivers performance comparable to real-world datasets.
>
> ## 2 Pretrained Model Ability
>
> **Human faces exhibit inherent diversity.** While judging realism can be challenging, we validate model capability using prompt fidelity and FID scores. CLIP scores in Table 2 of the main paper show that our method **performs comparably to real-world data and other approaches** on images from diverse scenes. FID scores in Table 8 of the appendix further indicate that, relative to prior work, the distribution of images generated by our approach **lies closer to that of the base generative model**.
>
> Additional examples in Figures 10, 11, and 12 in the appendix qualitatively demonstrate the generated image quality.
>
> ## 3 Base Generative Model
>
> We implement our method on SDXL to ensure **fair comparisons** with prior work. Consistory also uses SDXL, while StoryDiffusion builds on RealVisXL, a fine-tuned version of SDXL. Our approach trains only a lightweight mapping network for the SDXL text encoder, keeping all other parameters frozen. This design is **architecture-agnostic** and can potentially extend to other text-based generative models.
>
> ## 4 Contrastive Learning Effectiveness
>
> Contrastive learning forms the core mechanism for optimizing the MLP. Beyond demonstrating overall performance through face similarity, diversity, and prompt fidelity metrics, we further validate its effectiveness in an ablation study (Table 3 in the main paper), which verifies the contributions of positive and negative samples. The results confirm that **both positive and negative samples are essential** for optimal performance.
>
> ## 5 General Categories
>
> We evaluate our method on general categories, including animals and artificial objects. While performance does not yet match that on human portraits, the results show **promising consistency** and demonstrate the **broad applicability** of our approach.
>
> The core contribution of our work lies in the **lightweight, training-efficient framework**. Strong performance on human portraits validates the contrastive learning approach, while promising results on general categories demonstrate its **extensibility and potential**. This foundation provides valuable insights for future research in multi-category subject consistency.
>
> Large-scale evaluation with additional categories is left for the future work, but current results already establish CoCoIns as a **versatile and practical solution** for subject consistency across diverse domains.

---

> ### Author Response · Authors · 2025-08-15
> **Author Response (2/2)**
>
> ## 6 Interest for TMLR Audience
>
> Our work enables users to create **consistent characters across individual generations**, which is particularly beneficial for long-form content. Users can generate multiple distinct pieces and then combine them into a cohesive final product.
>
> As discussed in Section 2 (Subject-Driven Generation), subject-driven generation offers a similar solution but faces two key challenges. Tuning-based approaches such as DreamBooth are **time-consuming**, while encoder-based methods like PhotoMaker require specific designs to **integrate reference-encoding modules**, often necessitating additional training.
>
> Another approach, described in Section 2 (Subject-Consistent Generation), generates images in batches, allowing samples to access intermediate results and features from other samples. While this method avoids training, tuning, or additional modules, the assumption that all samples and features remain available for future generation can be **neither practical nor user-friendly**.
>
> We include the comparison table suggested by Reviewer pT3i to better illustrates the differences from other approaches and highlight the merits of CoCoIns.
>
> | Approach                      | Extra Modules       | Reference | Training       | Inference Constraints |
> |:----------------------------- |:------------------- |:--------- |:-------------- |:--------------------- |
> | Tuning-based Personalization  | None                | Yes       | Subject Tuning | None                  |
> | Encoder-based Personalization | Pretrained Encoders | Yes       | Once           | None                  |
> | Subject-Consistent Generation | None                | No        | None           | Batch                 |
> | **CoCoIns (Ours)**            | Lightweight MLP     | No        | Once           | None                  |
>
> The proposed approach is of interest to a wide range of content creators and practitioners. The methodologies and findings in this work can apply to numerous learning and vision tasks related to consistent and diverse image generation.
>
> We will revise and clarify these points in the updated paper.

---

### Review · Reviewer_pT3i · 2025-07-27

**Summary Of Contributions:**

Though existing text-to-image generative models follow the prompts and produce diverse images, they fall short on generating consistent appearances for the same subjects across multiple generations. This paper introduces Contrastive Concept Instantiation (CoCoIns) to improve the consistency of the same subjects across multiple individual generations. CoCoIns only learns a mapping function in a self-supervision setting to produce virtual word tokens which correspond to different subjects. The experimental results show the effectiveness of CoCoIns in terms of human subjects (quantitatively and qualitatively), and also demonstrate its potential on other subjects (e.g., dogs, cats) and on multiple-subject scenarios.

**Audience:**

Yes

**Audience Explanation:**

See above.

**Claims And Evidence:**

Yes

**Claims Explanation:**

**Strengths**
1. The learning overhead is low. CoCoIns only learns a mapping function, instead of tuning the text-to-image backbones. Plus, it does not require reference images/subjects due to its self-supervision nature.
2. The virtual word tokens correspond to language embeddings, which allows easy manipulation of subject appearances through text conditions.
3. The experimental results show that CoCoIns can generate images with consistent subjects over multiple generations in terms of humans, cats, dogs, and even multiple-subject scenarios.


**Weaknesses**
1. The negative prompts (same caption as the anchor prompt but inserted with a different latent code) can be considered as easy negative samples (or at least too different from anchor samples which may contribute to the large distance between anchor samples and negative samples at the beginning of training as mentioned in Section 4.1). Given the contrastive learning nature of CoCoIns, have the authors considered (1) increasing batch size (i.e., the number of negatives) and (2) making the negative samples more challenging (e.g., applying different amount of random noise to the anchor latent code).
2. There are a lack of ablation studies on (1) \lambda_con and \lambda_back and (2) the negative distance weight schedule hyperparameters of \gamma and \beta.
3. The demonstration of multi-subject consistency is interesting. However, there are no comparisons with other existing methods to demonstrate the performance of CoCoIns.

**Requested Changes:**

1. Can the authors add a table to show which prior works require generating images in batches or through reference tuning or need to tune the text-to-image backbones? This can better highlight the limitations of prior works and promote CoCoIns.

---

> ### Author Response · Authors · 2025-08-15
> **Author Response (1/2)**
>
> Thank you for the comments. We appreciate recognition of the low learning overhead and the ease of subject manipulation of CoCoIns. In response to the weaknesses (W) and requested changes (C), we provide the following discussion and revise the paper accordingly.
>
> We highlight updated sentences and tables in red in the revised paper.
>
> ## W1 Negative Samples
>
> Both ideas offer promising directions and warrant further exploration. We conduct experiments based on these ideas and present the results below.
>
> ### (1) Increasing negative samples
>
> This approach aligns with prior work in contrastive learning, which recommends large batch sizes containing many negative samples. In our current setup, each training example consists of an anchor, a positive, and a negative sample. The anchor and negative share the same noisy image and prompt but pair with different latent codes.
>
> To examine the effect of more negative samples, we generate additional latent codes and pair them with the same noisy image and prompt as the anchor. Negative distances in the contrastive loss are computed by averaging the distances from all negative samples.
>
> We compare the current setting (1 negative sample) with configurations using 4 and 8 negative samples. Table I shows the results. While 8 negative samples improve similarity, they reduce face diversity. The current setting achieves a better balance between these factors.
>
> Table I. Comparison of negative sample sizes.
>
> | Num Neg Samples | Sim   | Div   | CLIP  |
> | --------------- | ----- | ----- | ----- |
> | 1 (Current)     | 0.600 | 0.799 | 0.290 |
> | 4               | 0.547 | 0.697 | 0.291 |
> | 8               | 0.646 | 0.634 | 0.291 |
>
> ### (2) Applying noise to anchor latent codes
>
> As noted in Section 4.1, the model initially ignores randomly initialized latent codes, producing identical outputs for anchor and negative codes and resulting in large negative distances. To address this, we apply an increasing schedule to weight negative distances and stabilize training.
>
> Adding random noise to anchor latent codes provides an alternative way to control negative distances by adjusting the similarity between anchor and negative codes.
>
> We implement this idea by replacing negative codes with interpolations between anchor and randomly sampled negative codes, similar to generating negative codes by perturbing anchor codes. For each training triplet, we sample a ratio to interpolate between the two codes and normalize the negative distances by this ratio to prevent excessively large values.
>
> Table II compares the current setting with latent code perturbation. Although perturbation currently performs worse, it presents an interesting direction for future investigation.
>
> Table II. Comparison of sampling negative latent codes randomly versus perturbing anchor latent codes.
>
> | Neg Code          |  Sim  |  Div  | CLIP  |
> | ----------------- |:-----:|:-----:|:-----:|
> | Random (Current)  | 0.600 | 0.799 | 0.290 |
> | Perturbing Anchor | 0.508 | 0.642 | 0.290 |
>
>
> ## W2 Ablation Study
>
> We conduct ablation studies on (1) balancing consistency and background loss ($\lambda_\text{con}$, $\lambda_\text{back}$) and (2) scheduling the negative loss during training ($\gamma$, $\beta$).
>
> Table III shows results for varying $\lambda_\text{back}$ with $\lambda_\text{con}$ fixed at 1. Table IV shows results for different $\gamma$ values, and Table V presents comparisons across $\beta$ values. Consistent with previous observations, some hyperparameter choices yield high similarity but low diversity or prompt fidelity. We select the hyperparameters that **balance all key metrics**.
>
> These results and discussion are also included in Appendix Section C.2.
>
> Table III. Ablation study on weighting background loss.
>
> | $\lambda_\text{back}$ |  Sim  |  Div  | CLIP  |
> | --------------------- |:-----:|:-----:|:-----:|
> | 10                    | 0.637 | 0.603 | 0.265 |
> | 30                    | 0.600 | 0.799 | 0.290 |
> | 50                    | 0.516 | 0.707 | 0.292 |
>
> Table IV. Ablation study on weighting negative distances.
>
> | $\gamma$  |  Sim  |  Div  | CLIP  |
> | --------- |:-----:|:-----:|:-----:|
> | $10^{-4}$ | 0.666 | 0.766 | 0.288 |
> | $10^{-5}$ | 0.600 | 0.799 | 0.290 |
> | $10^{-6}$ | 0.445 | 0.635 | 0.293 |
>
> Table V. Ablation study on the schedule of negative distance weighting.
>
> | $\beta$ |  Sim  |  Div  | CLIP  |
> | ------- |:-----:|:-----:|:-----:|
> | 1       | 0.528 | 0.735 | 0.291 |
> | 2       | 0.600 | 0.799 | 0.290 |
> | 3       | 0.520 | 0.710 | 0.291 |

---

> ### Author Response · Authors · 2025-08-15
> **Author Response (2/2)**
>
> ## W3 Multi-Subject Comparison
>
> We update Figure 12 in the appendix to include comparisons with Consistory and StoryDiffusion using their public codebases. Although these codebases do not officially support multi-subject generation, we use prompts with identical subject terms ("a man and a woman"). Their approaches show some consistency, but our method achieves **higher image quality and consistency** while retaining flexibility by **allowing users to specify which subject words to align**.
>
> ## C1 Comparison of Settings and Implementation
>
> We create Table VI to compare our approach with previous work, as suggested, and include it in Appendix Section A.
>
> Table VI compares the **settings and implementation of our method** with those of prior work, categorized into subject-driven personalization and subject-consistent generation.
>
> Subject-driven approaches personalize a generative model by tuning parameters on reference images or incorporating additional pretrained encoders. These approaches are either time-consuming due to subject-specific tuning or require integration of general encoders such as DINO or domain-specific encoders like face recognition models.
>
> Subject-consistent approaches modify the prompt or attention mechanisms of the base generator, avoiding extra modules, reference images, or additional training. Their main limitation is the requirement for generating images in batches or reliance on stored features.
>
> Our method is more **lightweight and flexible**. It requires only an MLP trained once and supports individual inference.
>
> Table VI. Comparison of settings and implementation.
>
> | Approach                      | Extra Modules       | Reference | Training       | Inference Constraints |
> |:----------------------------- |:------------------- |:--------- |:-------------- |:--------------------- |
> | Tuning-based Personalization  | None                | Yes       | Subject Tuning | None                  |
> | Encoder-based Personalization | Pretrained Encoders | Yes       | Once           | None                  |
> | Subject-Consistent Generation | None                | No        | None           | Batch                 |
> | **CoCoIns (Ours)**            | Lightweight MLP     | No        | Once           | None                  |

---

### Review · Reviewer_HQRe · 2025-08-19

**Summary Of Contributions:**

## Summary
This work proposes Contrastive Concept Instantiation (CoCoIns), a lightweight framework for subject preservation in text-to-image generative models. The authors propose a simple and intuitive contrastive learning method by constructing positive and negative prompt pairs for a given subject and learning a mapping function from a sampled latent code to a pseudo-word as a learnable prefix to the subject token. Empirical results show the effectiveness of the proposed method in terms of subject preservation and diversity in generation.

## Strengths
- Overall, the paper is written very well with a clear motivation for developing the proposed CoCoIns method.
- The method is intuitive and proposes a strategy similar to "prefix tuning" or soft prompting that has been popularized in LLM/VLMs for a unique application in the subject preservation task.
- The experimental results are very convincing and show that CoCoIns produces high-quality generated images preserving subjects while maintaining diversity.
- The masked contrastive loss for the subject region in the image coupled with a background preservation loss is intuitive for the subject preservation task.

## Weaknesses
- Although the method is interesting, the applicability to long and arbitrary prompts is not particularly addressed. The experiments include a free-form prompt setting, but it is not clear how this method would work for complex subject relationships in prompts.
- The masking component may not always be accurate, especially when the subject can be ambiguous in images.
- It is not clear how the original latent code $\mathbf{z}$ is sampled.

**Audience:**

Yes

**Audience Explanation:**

This work is very clearly in line with the interests of TMLR's audience. With the popularity of text-to-image generative models, this paper makes a contribution to interpretability and model steering. Furthermore, the findings of this paper can be utilized for a variety of different applications in the generative modeling and computer vision domains.

**Claims And Evidence:**

Yes

**Claims Explanation:**

The claims made in this paper are accurate, convincing, and clear. The motivation behind developing this method comes from the computational cost of previous subject preservation methods. The proposed CoCoIns framework is intuitive in formulation and is reminiscent of "prefix tuning" or soft prompting that has been popularized in LLMs/VLMs. Using this idea for improving the deterministic components of text-to-image models is interesting and the empirical evaluations clearly show an improvement in generation quality compared to other methods.

**Requested Changes:**

Explaining the following in more details in the main text would be critical for clarity:
- How do you make sure that after the pseudo-word is prepended to the subject that the text encoder doesn't corrupt the embeddings? Is the assumption that the pseudo-word lies in the text embedding space? If so, how do you align this to make sure it is meaningful? The loss seems to be with respect to the generated images and a loss is not explicitly defined on the pseudo-words. Making this clear is important.
- How are the original latent codes sampled? Clarifying this is important since the method is largely dependent on transforming these codes to derive pseudo-words.
- How do you deal with subject ambiguity and more complex prompts?
- The out-of-the-box segmentation models are not always accurate for more complex images. How is the masked subject loss defined in such scenarios?

---

> ### Author Response · Authors · 2025-08-30
> **Author Response (1/2)**
>
> Thank you for the review and feedback. We appreciate you highlighting the clear motivation, the intuitive design of our method, and the convincing, high-quality results from our experiments. We address the raised issues here and include the requested changes in the revised manuscript highlighted in red.
>
> ## W1 Complex Scenarios
>
> In this work, we focus on developing a method for synthesizing consistent images via instantiated concepts and demonstrate results on faces as they are arguably the most interesting objects in this field. In addition, we show some results on other objects such as dogs, cats, and cars (see Figure 6). These results show that our method can be applied to generic objects.
>
> **Long, Arbitrary Prompts**: To evaluate our method beyond simple portrait descriptions, we prepare the Scenes test set. These prompts, generated by an LLM, describe subjects in a wide range of real-world contexts and introduce significantly more complexity. Figure 4 and 11 of the manuscript shows that CoCoIns performs favorably in this setting, achieving strong subject similarity while maintaining high diversity and prompt fidelity. This demonstrates that our method can successfully maintain a subject's identity in more descriptive and varied narratives.
>
> **Complex Subject Relationships**: We also evaluate the model's potential to handle multiple subjects in a single prompt. Figure 7 and 12 of the manuscript shows that CoCoIns successfully generates images with two distinct and consistent subjects by providing two different latent codes, even though the model is only trained on single-subject images. We are encouraged to see that the model maintains consistency for both individuals in the same scene. These results indicate a strong potential to extend our framework to more complex scenarios with multiple characters and objects.
>
> We see this work as a critical first step to synthesize images of consistent subjects. We will extend CoCoIns along the suggested directions in our future work.
>
> ## W2 and C3 Subject Ambiguity
>
> Our experiments focus on single-subject consistency. To create a clean training dataset and minimize ambiguity, we use CelebA, which primarily contains human portraits. We generate masks using a powerful, off-the-shelf model, Grounding SAM 2. Then we manually filter out images with more than one face according to segmentation results. This step ensures that the training images contain single subjects and reduces ambiguous masks.
>
> As discussed above, although the model is trained only on single-subject portrait images, it can handle subject consistency in more challenging conditions, such as free-form prompts in _Scenes_ (Figure 4 and 11) and images with two subjects (Figure 7 and Figure 12).
>
> ## W2 and C4 Segmentation Models
>
> In the single-subject experiments, the issue of inaccurate segmentation is less pronounced. The task of segmenting a person in a portrait, as is common in the CelebA dataset, is relatively straightforward for a powerful model like Grounded SAM 2. In addition, we have manually validated 100 randomly sampled images and find the predictions to be consistently reliable for this use case.
>
> For more challenging scenarios with multi-subject images, the proposed loss function framework can be further extended. Since the prior work [A] has shown the relationships between word embeddings and feature maps, and a pseudo-word functions in the text embedding space to describe the subject that follows, we hypothesize that the model can learn to associate a  pseudo-word with features of its corresponding subject, even in a multi-subject scene.
>
> Nevertheless, we agree that this is a hypothesis that requires more thorough experimentations and evaluations, which will be part of our future work.
>
> [A] Amir Hertz, Ron Mokady, Jay Tenenbaum, Kfir Aberman, Yael Pritch, Daniel Cohen-Or. Prompt-to-Prompt Image Editing with Cross-Attention Control. arXiv:2208.01626.
>
>
> ## W3 and C2 Latent Code Sampling
>
> Latent codes are randomly sampled from Gaussian Distribution $N(0, I)$, stated in Equation 3. In each training triplet, the anchor and positive samples share the same latent code, and the negative sample is paired with another randomly sampled code, as detailed in Equation 5.
>
> ## C1 Pseudo-Words
>
> ### 1 Preventing Prompt Corruption
>
> To ensure a pseudo-word only affects the subject and does not _corrupt_ the rest of the prompt, we use a **background preservation loss**. This loss minimizes the difference in the background areas between an image generated  with the pseudo-word and one generated without it. This localizes the effect of a pseudo-word to the masked subject area, thereby preserving the overall scene context dictated by the original prompt. The ablation study in Table 4 shows that removing this loss significantly harms prompt fidelity.

---

> > ### Author Response · Authors · 2025-08-30
> > **Author Response (2/2)**
> >
> > ### 2 The Place of Pseudo-Words
> >
> > This is correct. The core assumption is that the pseudo-word lies in the text embedding space. Our mapping network $f$ is explicitly designed to take a latent code $\boldsymbol{z}$ and output a pseudo-word $\boldsymbol{w}$ as a vector that has the same dimension as the text embeddings of the pre-trained text encoder. This pseudo-word vector is then inserted directly into the sequence of prompt embeddings before the subject token.
> >
> > ### 3 Alignment and Loss Definition
> >
> > The loss is not defined on the pseudo-words directly. Instead, a pseudo-word is made meaningful through an indirect alignment process using a contrastive loss on the generated images.
> >
> > We use a triplet loss that operates on the predicted image latents from the denoiser. The model is trained to minimize the visual difference between images generated with the same pseudo-word (even with different prompts and noise) while maximizing the visual difference between images generated with different pseudo-words. This process forces the model to treat each pseudo-word as a unique identifier for a specific visual appearance. The _meaning_ of a pseudo-word becomes the consistent subject identity it produces.
> >
> > We also analyze the space of pseudo-words in Appendix C.3, which provides strong evidence that this indirect alignment is successful. The experimental results show that:
> > * **Interpolation**: Smoothly interpolating between two latent codes results in a smooth visual transition between the two corresponding faces.
> > * **Similarity**: Pseudo-words that are closer to each other in the embedding space (i.e., have higher cosine similarity) produce subjects with a more similar appearance.
> >
> > These results demonstrate that the learned space of pseudo-words is well-structured and meaningful.

---

### Decision · Action_Editor_e6ik · 2025-10-22

**Recommendation:** Accept as is

**Additional Comments:**

The AE reviewed all the final comments from the reviewers.

All the reviewers acknowledged the clarity and technical soundness of the proposed method.

However, no major concerns were raised.

One reviewer encouraged the authors to add additional experiments and visualizations to make the work more complete—particularly quantitative evaluations on multi-subject generation and studies using more advanced diffusion models. Nevertheless, even without these evaluations, the original claims appear to be well supported, and the submission is considered suitable for acceptance in TMLR.

**Audience:**

Yes

**Audience Explanation:**

Its insights into prompt-based "subject consistency" would interest researchers working on text-to-image generation, controllable generation, and efficient model adaptation.

**Claims And Evidence:**

Yes

**Claims Explanation:**

The paper presents clear technical motivation, well-designed methodology, and solid experimental evidence supporting its claims. The authors provided sufficient clarifications and additional experiments during rebuttal, addressing earlier concerns.

Overall, the evidence convincingly demonstrates the effectiveness of CoCoIns for subject preservation in text-to-image generation.